# X-VILA: Cross-Modality Alignment for Large Language Models

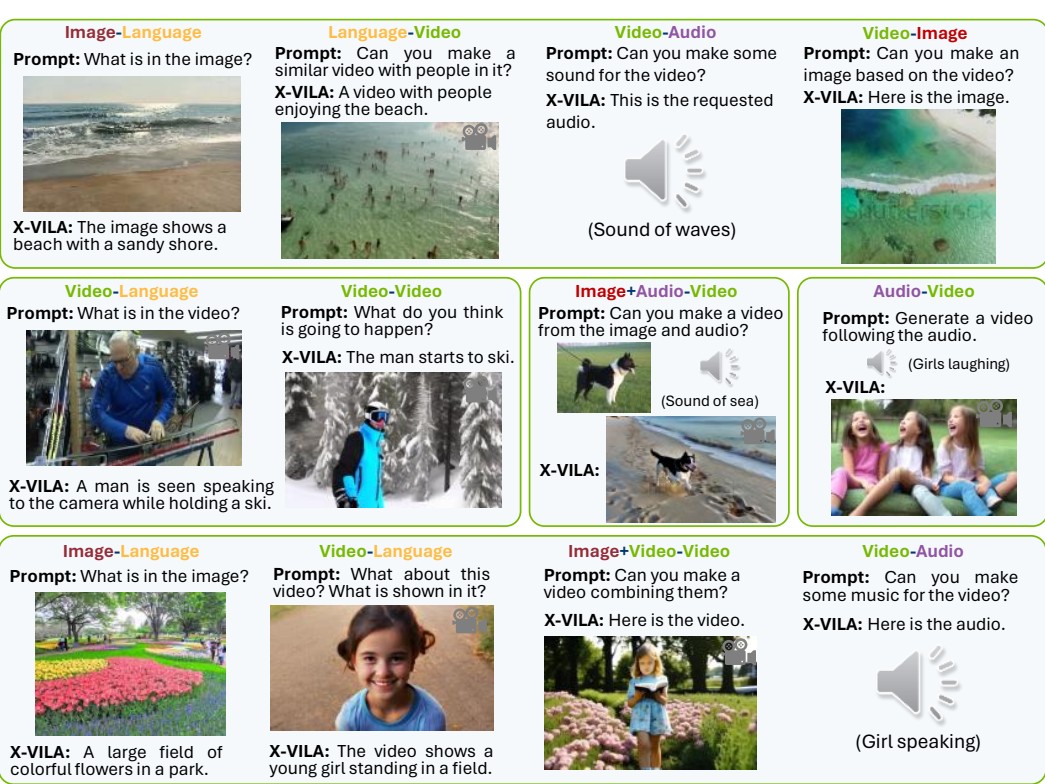

Figure 1: We introduce **X-VILA**, a foundation model for **cross-modality** understanding, reasoning, and generation in the domains of **video**, **image**, **language**, and **audio**.

## Abstract

We introduce X-VILA, an omni-modality model designed to extend the capabilities of large language models (LLMs) by incorporating image, video, and audio modalities. By aligning modality-specific encoders with LLM inputs and diffusion decoders with LLM outputs, X-VILA achieves cross-modality understanding, reasoning, and generation. To facilitate this cross-modality alignment, we curate an effective interleaved any-to-any modality instruction-following dataset. Furthermore, we identify a significant problem with the current cross-modality alignment method, which results in visual information loss. To address the issue, we propose a visual alignment mechanism with a visual embedding highway module. We then introduce a resource-efficient recipe for training X-VILA, that exhibits proficiency in any-to-any modality conversation, surpassing previous approaches by large margins. X-VILA also showcases emergent properties across modalities even in the absence of similar training data. The project will be made open-source.

## 1 Introduction

Large language models (LLMs) provide an emerging foundation for enhancing various deep learning tasks beyond the realm of natural language processing. As an example, research community has

been quickly extending the fast progress of LLMs (Devlin et al., 2019; Raffel et al., 2020; Dai et al., 2019; OpenAI, 2023b; Touvron et al., 2023a;b; Taori et al., 2023; Chiang et al., 2023; Karamcheti et al., 2021; Penedo et al., 2023; Chowdhery et al., 2022; yi, 2023; Bai et al., 2023a) towards the computer vision (CV) domain (Liu et al., 2024; Alayrac et al., 2022; Driess et al., 2023b; Chen et al., 2023; Li et al., 2023a; fuy, 2023; Bai et al., 2023b; OpenAI, 2023a; Zhu et al., 2023a). The introduction of LLMs in CV tasks enables vision models to perform many zero/few-shot and in-context learning tasks that are "promptable" through user questions, potentially empowering reasoning capabilities for the first time. Despite remarkable progress, cross-modality alignment is still a challenging task as the joint training stage for cross-modality learning requires carefully designed feedback signal (Wei et al., 2021; Dai et al., 2023) to guide the connected foundation models (Alayrac et al., 2022; Liu et al., 2024; Li et al., 2023a), backed by cross-modality datasets at scale (Zhu et al., 2023b; Byeon et al., 2022; Schuhmann et al., 2022). Hence, the majority of existing studies revolve around a solitary input modality linked to LLMs, with the output being solely text. For example, Flamingo (Alayrac et al., 2022), LLaVA (Liu et al., 2024), and VILA (Lin et al., 2024) delve into image input, while Video-LLaMA (Zhang et al., 2023a) and LITA (Huang et al., 2024) specifically concentrates on video input. Exploring the integration of various modalities into a cohesive framework is a crucial yet relatively unexplored research challenge (Tang et al., 2023; Wu et al., 2023; Lu et al., 2022) in the domain of multi-modality LLMs.

This study focuses on the development of a systematic approach to integrate multiple modalities, such as video, image, and audio, into an LLM at both the input and output stages. Our objective is to facilitate cross-modal conversations in an any-to-any modality (or "X-to-X") LLM, allowing for generation in different modalities. To accomplish the ambitious objective, we present a two-phase alignment mechanism: *(i) Textual alignment.* We align input and output representations of different modalities to the textual embedding space of the LLM (Wu et al., 2023). Specifically, in regard to the input of LLM, we use a unified embedding space that allows for the sharing of features extracted from encoders across diverse modalities. As for the output of LLM, we employ fine-tunable modality-specific diffusion models to convert the generated outputs of the LLM into content that aligns with the respective modalities. *(ii) Visual alignment.* We observe that the previous textual alignment alone fails to preserve visual features adequately in vision-to-vision generation tasks, such as image-to-video and video-to-image generation. This limitation can be attributed to the loss of information during the projection process from visual encoders to the LLM, as well as the LLM's tendency to prioritize common concepts over specific visual details. To address this issue, we propose a new module named Visual Embedding Highway (VEH). The VEH module facilitates the direct guidance of visual decoders by enabling visual features to bypass the LLM. By incorporating VEH, we have observed a notable enhancement in the correspondence of visual content between the input and output stages of our framework.

On the other hand, the scarcity of cross-modality instruction-following data poses a significant challenge in the development of any-to-any modality (or "X-to-X") LLMs. This limitation severely restricts the progress in creating LLMs that can seamlessly handle multiple modalities in both input and output ends. Existing datasets provide limited data, mostly in the form of X-to-text or text-to-X. Therefore, we curate a large-scale X-to-X dataset to facilitate cross-modality interactions between text, audio, image, and video. Overall, we synthesize more than 1.5M multi-modality conversations, with each conversation containing at least one cross-modality question-and-answer pair. This dataset has proven effective in our experiments for training models to achieve any-to-any modality capabilities.

To achieve the cross-modality input-output alignment of LLMs in our X-to-X LLM, we design three major training phases: **(i)** A data-effective alignment phase that involves aligning the multi-modality encoders with the LLM inputs and the multi-modality decoders with the LLM outputs. **(ii)** An interleaved multi-modality pre-training phase with interleaved instruction data across modalities for enhanced in-context learning performance. **(iii)** An X-to-X cross-modality instruction tuning phase that includes a two-step alignment process: textual alignment and visual alignment mechanism. Through our innovative approach to multi-modality alignment, we build a powerful X-to-X multi-modality LLM with the ability to comprehend and generate multi-modality content. We term our new model "X-VILA" for **cross-modality** understanding, reasoning, and generation in the domains of **V**ideo, **I**mage, **L**anguage, and **A**udio. For instance, as shown in Figure 1 and Figure 9, X-VILA demonstrates its capacity to recognize the subjects in the image, which results from our vision-language alignment training. Then, it can retrieve its knowledge and make logical deductions to

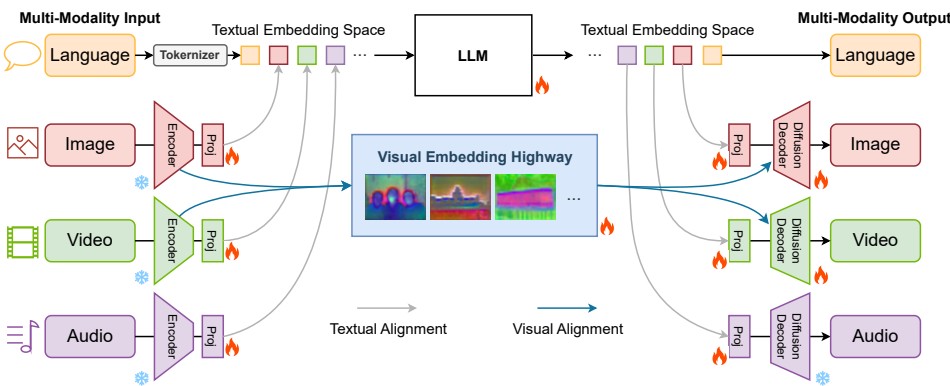

Figure 2: X-VILA schematic diagram. X-VILA augments a pretrained LLM towards new modalities via (i) connecting pretrained encoders to LLM input textual embedding space and (ii) connecting pretrained diffusion decoders to the LLM output textual embedding space (Section 2.1). The system is jointly trained via a new cross-modality alignment procedure (Section A).

answer the user's questions about the content in the image. Last but not least, it can generate aligned multi-modality output that matches the given context.

In summary, this work makes contributions in three aspects:

- A new family of any-to-any modality chat LLM that is able to conduct multi-modality conversations by understanding signals from different modalities and generating content in various formats, including video, audio, image, and text.

- A novel 2-step alignment mechanism that effectively aligns both semantic and visual details between the input and output spaces. This mechanism ensures a comprehensive and accurate correspondence between the input and output of our X-to-X LLM.

- The creation of a new X-to-X multi-modality instruction tuning dataset that is proven effective for cross-modality alignment. This dataset serves as a valuable resource for future research in the realm of multi-modality foundation models.

## 2 METHODOLOGY

### 2.1 X-VILA ARCHITECTURE

We consider four common modalities in this work: text, image, video, and audio. The tenet of X-VILA is an alignment-based architecture to augment an LLM with the ability to "see/hear/read" multi-modality inputs and "draw/speak/write" multi-modality outputs, as shown in Figure 2.

**Modality-specific encoders.** We adopt modality-specific encoders to handle inputs from different modalities. This strategy harvests the pre-trained understanding ability of the domain expert encoders and has been proven successful in many vision-language models (Alayrac et al., 2022; Li et al., 2023a; Liu et al., 2024). To better align embeddings of different modalities, we use ImageBind encoders (Girdhar et al., 2023), which unify features from different modalities, including image, video, and audio, into one feature space. Technically, for each modality $m \in \{$'text', 'image', 'video', 'audio'$\}$, we notate the encoders as $\mathbf{Enc}_m$. For text modality, the encoder is a text tokenizer (Kudo & Richardson, 2018), while for other modalities they are ImageBind transformers (Girdhar et al., 2023). We then use modality-specific trainable linear layers, notated as $\mathbf{P}_m^{\text{in}}$, to project $\mathbf{Enc}_m$ output into embedding sequences $\mathbf{S}$ in the textual embedding space of the following LLM. We can formulate this process as:

$$\mathbf{S}^{\text{in}} = \{\mathbf{P}_m^{\text{in}}(\mathbf{Enc}_m(\mathbf{X}_m))\}, \tag{1}$$

where $\mathbf{X}_m$ is input from different modalities $m \in \{$'text', 'image', 'video', 'audio'$\}$.

**Large language model (LLM).** LLM serves as the "brain" of our framework. It processes information from the textual embedding space and predicts language outputs correspondingly. We adopt Vicuna-7B-1.5 (Chiang et al., 2023; Touvron et al., 2023b), which demonstrates state-of-the-art language understanding and generation ability. For easier understanding, we slightly abuse the annotation and write the autoregressive process of generating output embedding sequence $\mathbf{S}^{\text{out}}$ by the LLM as:

$$\mathbf{S}^{\text{out}} = \mathbf{LLM}(\mathbf{S}^{\text{in}}). \tag{2}$$

**Modality-specific decoders.** For generating multi-modality outputs other than text, we adopt the "modality-specific generation tokens" designed by (Wu et al., 2023). Other than common text tokens, there are three types of modality-specific generation tokens: image generation tokens $\{[\text{IMG}_i], i \in [1, N_{img}]\}$, video generation tokens $\{[\text{VID}_i], i \in [1, N_{vid}]\}$, and audio generation tokens $\{[\text{AUD}_i], i \in [1, N_{aud}]\}$. $N_{img}$, $N_{vid}$, and $N_{aud}$ are the numbers of generation tokens for image, video, and audio, respectively. These modality-specific generation tokens are added to the vocabulary of LLM. The LLM is trained to predict when to generate these modality-specific generation tokens, and these generation tokens will be translated for the synthesis of image, video, or audio, via a set of modality-specific decoders (*i.e.*, generation models). Technically, we extract the subset of output embedding sequence $\mathbf{S}^{\text{out}}$ corresponding to the aforementioned generation tokens of modality

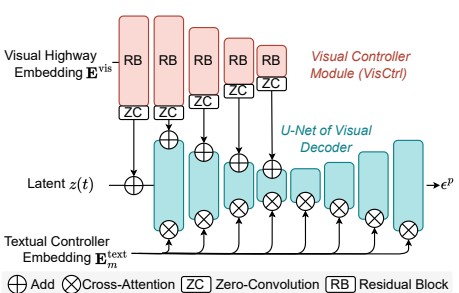

Figure 3: Illustration of the proposed visual embedding highway in X-VILA. The visual highway embedding $\mathbf{E}^{\text{vis}}$ is obtained from the visual encoder. The design incorporates a visual controller module responsible for processing $\mathbf{E}^{\text{vis}}$ and generating control signals. These signals are then incorporated into various layers of the U-Net in visual decoders. $\mathbf{E}_m^{\text{text}}$ is "textual controller embedding", which is the subset of output embedding sequence $\mathbf{S}^{\text{out}}$ corresponding to the generation tokens of modality $m$. $z(t)$ is the latent at reverse step $t$. $\epsilon^p$ is the predicted noise by U-Net.

$m$. We name this subset the generation embedding sequence $\mathbf{S}_m^{\text{gen}}$. We use modality-specific transformer layers, denoted as output projection layers $\mathbf{P}_m^{\text{out}}$, to project $\mathbf{S}_m^{\text{gen}}$ to the feature space of the original pre-trained text encoder of the modality-specific decoder. As the resulting embedding will be used to control the modality-specific decoder via cross-attention, we name the resulting embedding vector as "textual controller embedding" $\mathbf{E}_m^{\text{text}}$. Thus we have:

$$\mathbf{E}_m^{\text{text}} = \mathbf{P}_m^{\text{out}}(\mathbf{S}_m^{\text{gen}}). \tag{3}$$

(Wu et al., 2023) freezes the decoder models and only supervises the $\mathbf{E}_m^{\text{text}}$ to be similar to the original text encoders of the diffusion models. This behavior largely limits the synergy between generation models and the other parts of the model, as the learning target is essentially to mimic the pre-trained text encoder of the diffusion models. Differently, we include the modality-specific decoder models in fine-tuning to better align them with the LLM and other parts of the unified generative framework. The training details will be discussed in a later section. Specifically, to achieve a better multi-modality generation ability, we employ state-of-the-art generation models trained on large-scale data as modality-specific decoders. We adopt VideoCrafter2 (Chen et al., 2024) for video generation, Stable Diffusion 1.5 (Rombach et al., 2022) for image generation, and AudioLDM (Liu et al., 2023a) for audio generation.

**Visual embedding highway.** The weakness of the previously introduced text-space-based alignment is the inadequate visual features available at the output end, as can be seen in examples in Figure 5. Intuitively, this stems from the one-to-many correspondence between text and visual semantic spaces, *e.g.*, "city view" may relate to images varying in illumination and layout.

To address this issue, we propose a visual embedding highway that bridges the visual encoders and decoders, built to alleviate the information loss when projecting high-dimensional visual content to the textual embedding space. Specifically, we obtain the layer-wise feature maps from the Image-Bind visual encoder and add up these features as visual highway embedding $\mathbf{E}^{\text{vis}}$. $\mathbf{E}^{\text{vis}}$ has shape $H \times W \times C$, where $H$ and $W$ are height and width of the feature maps, $C$ is the embedding vector.

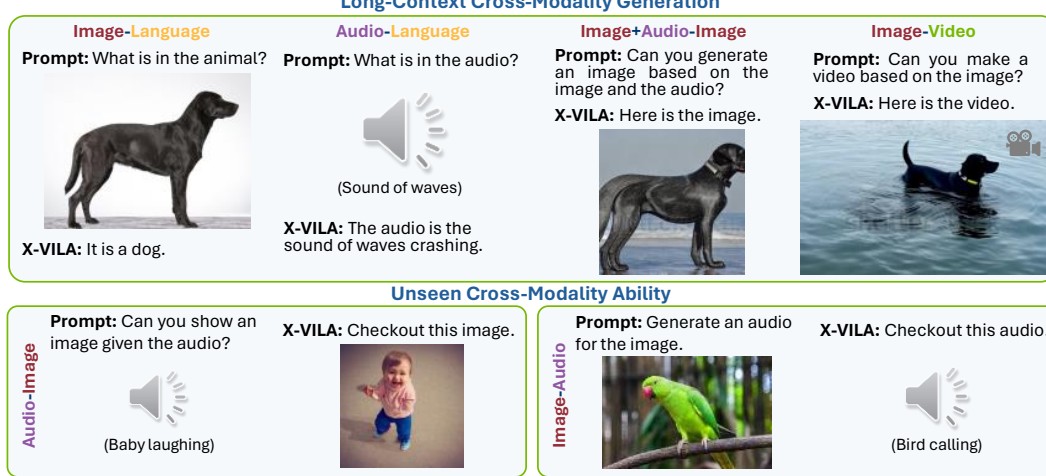

Figure 4: We observe emergent abilities of X-VILA without training on similar data: **(i) Long-context cross-modality generation ability.** Combine multiple inputs from different modalities and generate consistent content. **(ii) New types of cross-modality ability.** Conduct image-to-audio and audio-to-image generation tasks. Conversations are continuous left-to-right within each green box.

To control the decoder using $\mathbf{E}^{\text{vis}}$, we design a light-weight visual controller (VisCtrl) module based on the philosophy of (Mou et al., 2023; Zhang et al., 2023b) to process $\mathbf{E}^{\text{vis}}$. The controller module comprises 4 stages, where each stage consists of two residual convolutional blocks. These blocks have cascading spatial dimensions that match the resolution settings in the U-Net encoder (Rombach et al., 2022) of image/video decoders. In each stage, there is an additional convolutional block initialized with zero weights. This block generates output control signals for the stage, which are initially zero at the start of the training. These control signals are added to different stages of the U-Net, as shown in Figure 3. Inspired by (Xiao et al., 2023), we employ a conditioning rate $\alpha \in [0, 1]$ to regulate the proportion of steps conditioned by visual features. Therefore, the noise prediction process in each reverse step $t$ in the visual decoders can be written as:

$$\epsilon^p = \begin{cases} \text{U-Net}_m(z(t), \text{VisCtrl}_m(\mathbf{E}^{\text{vis}}), \mathbf{E}_m^{\text{text}}) & \text{if } t < T \times \alpha \\ \text{U-Net}_m(z(t), \text{Null}, \mathbf{E}_m^{\text{text}}) & \text{if } t \geq T \times \alpha \end{cases}, m \in \{\text{`image', `video'}\}. \quad (4)$$

where $\epsilon^p$ is the predicted noise given input latent $z(t)$, $T$ is the number of diffusion steps, U-Net$_m$ is the U-Net of the diffusion decoder for modality $m$, and VisCtrl$_m$ is the visual control module for modality $m$. "Null" means no VEH feature is passed to the U-Net at the corresponding timestep. During instruction tuning process on X-to-X datasets, both the U-Net and the controller modules are fine-tuned together. This manner ensures a better synergy between decoders and the LLM.

The experimental results introduced in the later sections show that the proposed visual embedding highway can significantly increase the consistency between the generation results and the visual context of our multi-modality unified generation model.

## 2.2 X-VILA TRAINING

The training process of X-VILA is divided into three phases, namely (i) encoder-LLM-Decoder alignment training, (ii) interleaved data pre-training, and (iii) X-to-X cross-modality instruction fine-tuning. We describe the details of X-VILA training in Appendix A due to space limit.

## 3 EXPERIMENTS

### 3.1 DATASETS AND EVALUATION

**Setup.** In this work, we utilize different datasets for different training phases. For the first encoder-LLM-decoder alignment training, the X-text pairs from academic datasets as in prior work of (Liu et al., 2024; Sharma et al., 2018; Bain et al., 2021; Kim et al., 2019; Mei et al., 2023). During

Table 1: Statistics of the any-to-any modality SFT dataset MosIT (Wu et al., 2023) and ours. Our X-to-X dataset has a significantly larger data volume. We will open source the dataset to the academic community.

| Dataset | Total | IMG2VID | VID2IMG | VID2VID | AUD2VID | VID2AUD | IMG+AUD2VID |
|---|---|---|---|---|---|---|---|
| MosIT dataset | 5K | - | - | - | - | - | - |
| X-to-X dataset (ours) | 1.6M | 509,924 | 509,924 | 509,924 | 32,874 | 32,874 | 32,874 |

the interleaved data pre-training phase, we construct interleaved multi-modality corpus from (Zhu et al., 2023b; Krishna et al., 2017). Overall our X-text training stage contains 12M pairs in total, and our interleaved corpus contains 1M samples. We will open source our datasets for the academic community.

In the final X-to-X cross-modality instruction tuning, we create **a new X-to-X dataset** to enhance cross-modality alignment. We synthesize conversation samples in 6 types based on the modalities in input and output ends: video-to-image, video-to-video, image-to-video, video-to-audio, audio-to-video, image&audio-to-video. Statistically, we construct 0.5M image-to-video, 0.5M video-to-image, 0.5M video-to-video, 32 audio-to-video, 32K video-to-audio, and 32K image+audio-to-video conversations. Each conversation contains more than one pair of cross-modality Q&A pairs. The overall statistics are shown in Table 1. Some conversation examples are shown in Figure 10. We blend our X-to-X dataset with SFT datasets from LLaVA (Liu et al., 2024), VideoChat (Li et al., 2023b), NextGPT-instructions (Wu et al., 2023), and Alpaca (Taori et al., 2023).

**Evaluation.** For benchmarking the X-to-X alignment ability of different models, we randomly curate a validation subset from (Bain et al., 2021) and (Krishna et al., 2017) to build the cross-modality conversations for evaluation. Overall the evaluation set contains 200 video-to-image, 200 image-to-video, 200 video-to-video, 62 audio-to-video, 62 image+Audio-to-video, and 62 audio-to-video conversations for evaluation. We will also open source the validation benchmark for academic community. In order to evaluate the similarity between ground-truth annotations and model predictions across different modalities, we introduce a metric called the "X-to-X Alignment Score ($X^2A$ Score)". To compute this score, we employ the ImageBind transformer (Girdhar et al., 2023) to extract embedding vectors from the audio, video, and image predictions as well as the corresponding ground truths. We then calculate the cosine similarity scores between these vectors. The resulting scores are presented as percentages, ranging from 0 to 100. Finally, we average the scores across all validation samples to obtain the $X^2A$ scores for each type of data.

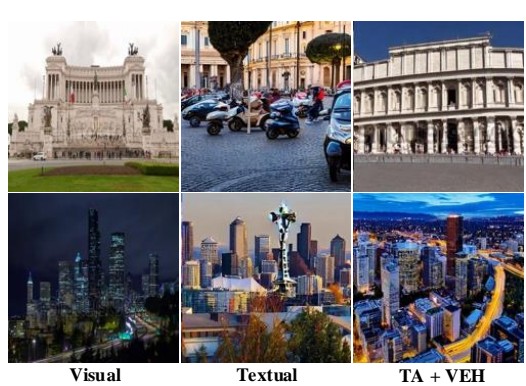

**Visual Reference**    **Textual Alignment (TA)**    **TA + VEH (X-VILA)**

Figure 5: Effectiveness of the proposed visual embedding highway network. Given the visual reference image/video, we prompt the model with "`Please generate an image similar to the semantics in the input.`" Compared to textual alignment only (TA), our visual embedding highway (VEH) helps preserve visual details from the visual inputs.

**Baseline methods.** We conduct a comparison between our model and Next-GPT (Wu et al., 2023), a recently introduced instruction-following LLM designed for multi-modality understanding and generation. Their method is restricted to textual alignment exclusively.

## 3.2 QUANTITATIVE ANALYSIS AND ABLATION STUDY

**Effectiveness of Visual Embedding Highway.** We compute the aforementioned $X^2A$ scores of different models on the X-to-X alignment benchmarks built upon (Krishna et al., 2017) and (Bain et al., 2021), and present the results in Table 2 and Table 3 separately. Specifically, we study the $X^2A$ scores of Next-GPT and different versions of our X-VILA model. We investigate the performance of our model under different scenarios: **(i)** utilizing only textual alignment, **(ii)** incorporating visual

Table 2: Ablations on X-VILA training strategies with reference to prior work for visual feature enhancements. "w/ X2X text" denotes using our X-to-X dataset for textual alignment only. "VEH (img)" denotes using the proposed visual embedding highway (VEH) for image decoder, while "VEH (img+vid)" denotes using VEH for both image and video decoders. We observe that image generation task is significantly improved after using VEH (img), and the video generation tasks are boosted after using VEH on video decoder.

| Method | VID2IMG ($\uparrow$) | VID2VID ($\uparrow$) | IMG2VID ($\uparrow$) |
|---|---|---|---|
| Next-GPT (Wu et al., 2023) *ICML'24* | 27.85 | 10.47 | 13.08 |
| **X-VILA** w/ X2X text | 36.09 | 46.18 | 45.93 |
| **X-VILA** w/ X2X text + VEH (img) | **44.06** | 46.68 | 45.94 |
| **X-VILA** w/ X2X text + VEH (img+vid) – *final design* | 43.95 | **49.76** | **48.81** |

Table 3: Ablations on X-VILA training strategies with reference to prior work considering all modalities for inputs and outputs. "w/ X2X text" denotes using our X-to-X dataset for textual alignment only. "VEH (img)" denotes using the proposed visual embedding highway (VEH) for image decoder, while "VEH (img+vid)" denotes using VEH for both image and video decoders. The effectiveness of visual embedding highway is solid for image and video generation.

| Method | VID2IMG ($\uparrow$) | IMG+AUD2VID ($\uparrow$) | VID2AUD ($\uparrow$) | IMG2VID ($\uparrow$) | VID2VID ($\uparrow$) | AUD2VID ($\uparrow$) |
|---|---|---|---|---|---|---|
| Next-GPT (Wu et al., 2023) *ICML'24* | 15.31 | 44.63 | 8.17 | 38.23 | 31.81 | 37.13 |
| **X-VILA** w/ X2X text | 53.82 | 49.54 | 22.79 | 42.94 | 44.42 | 42.23 |
| **X-VILA** w/ X2X text + VEH (img) | 67.40 | 48.64 | 23.53 | 42.66 | 43.04 | 42.04 |
| **X-VILA** w/ X2X text + VEH (img+vid) – *final design* | **67.94** | **59.71** | **23.87** | **57.01** | **57.39** | **49.44** |

alignment through the proposed visual embedding highway (VEH) on the image decoder, and **(iii)** extending VEH to both the image and video decoders.

Our findings indicate that even by utilizing textual alignment alone with our carefully curated X-to-X datasets, our model demonstrates a substantial performance advantage over Next-GPT. Moreover, as we progressively introduce the visual embedding highway to the image and video decoders, we observe consistent and significant improvements in visual understanding and generation tasks. In summary, our X-VILA demonstrates significantly stronger cross-modality understanding, reasoning, and generation ability on all types of conversation data. These results suggest the effectiveness of our X-to-X alignment strategy and the proposed visual embedding highway design. Notably, both Next-GPT and X-VILA are based on the ImageBind model, making it fair to use ImageBind scores for both models.

Table 4: X-VILA demonstrates comparable performance to domain experts when evaluated on targeted sub-modality tasks image-to-text benchmarks.

| Method | VQAv2 ($\uparrow$) | VisWiz ($\uparrow$) | MMMU-val ($\uparrow$) |
|---|---|---|---|
| BLIP-2 13B (Li et al., 2022) | 65.0 | 19.6 | - |
| InstructBLIP 13B (Dai et al., 2023) | - | 33.4 | - |
| Qwen-VL-Chat 7B (Bai et al., 2023b) | 78.2 | 38.9 | 35.9 |
| LLaVA 1.5 7B (Liu et al., 2023b) | **78.5** | 50.0 | **36.4** |
| **X-VILA 7B (ours)** | 72.9 | **50.9** | 33.9 |

Table 5: Extra comparison on audio and video benchmarks with AudioCaps (audio) and MSRVTT (video) validation sets.

| Method | Audio SPIDEr ($\uparrow$) | Audio CIDEr ($\uparrow$) | Video METEOR ($\uparrow$) |
|---|---|---|---|
| Next-GPT Wu et al. (2023) | 10.13 | 14.53 | 19.60 |
| **X-VILA 7B (ours)** | **12.99** | **16.61** | **22.49** |

**Influence of conditioning rates.** We present the X$^2$A scores plotted with varying conditioning rates $\alpha$ (Equation 4) in VEH (image), as depicted in Figure 8. Our observations indicate that an increase in $\alpha$, corresponding to more reverse steps exposed to VEH features during image sampling, leads to improved multi-modality alignment. This outcome aligns with our intuitive expectations.

**Extra multi-modality benchmarks.** To further evaluate the multi-modality understanding capabilities of X-VILA, we perform zero-shot experiments on several multi-modality VQA benchmarks, including VQAv2 (Goyal et al., 2017), VisWiz (Gurari et al., 2018), and MMMU-val (Yue et al., 2024). The results in Table 4 indicate that X-VILA is competitive with the leading domain-expert VLMs, while possessing the X-to-X capability. We also compare the performance with Next-GPT (Wu et al., 2023) on the audio understanding task using the AudioCaps validation split, as well as on the video understanding task using the MSRVTT validation set in Table 5. X-VILA demonstrates significantly better multi-modality understanding ability.

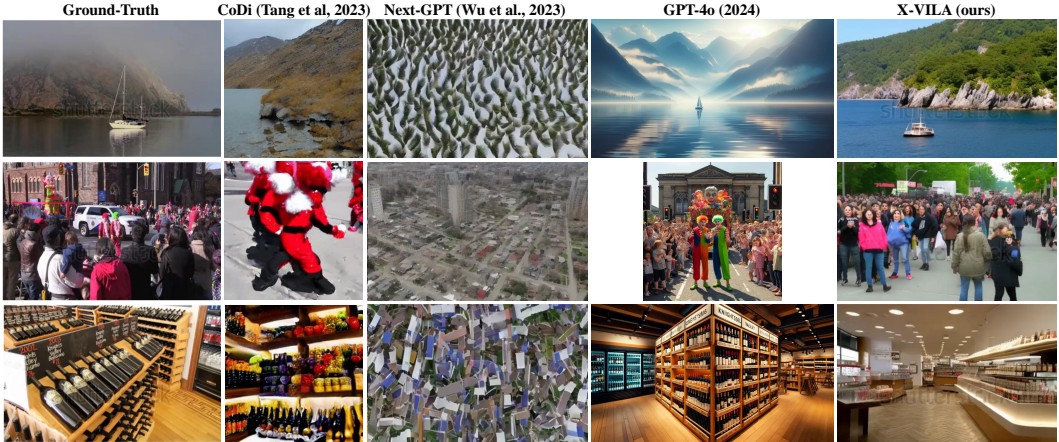

Figure 6: Visual comparison to the recent any-to-any modality LLMs including Next-GPT (Wu et al., 2023), CoDi (Tang et al., 2023), and GPT-4o (OpenAI, 2024) on the cross-modality alignment task to generate a video similar to the input image context. X-VILA demonstrates good generation quality and better visual cross-modality consistency. GPT-4o is only able to generate images but not videos.

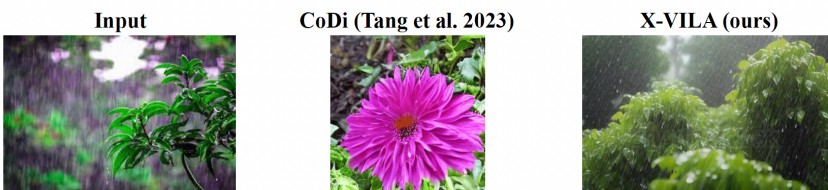

Figure 7: Visual comparison to the recent work CoDi (Tang et al., 2023) on cross-modality alignment for image input to video generation task. X-VILA demonstrates largely improved generation quality and cross-modality consistency.

### 3.3  QUALITATIVE ANALYSIS AND ABLATION STUDY

**Qualitative X-to-X alignment measurement.** We provide a qualitative comparison to the state-of-the-art any-to-any LLMs, namely Next-GPT (Wu et al., 2023), CoDi (Tang et al., 2023), and GPT-4o (OpenAI, 2024) on visual cross-modality alignment tasks in Figure 6 and Figure 7. We assess their performance by supplying an image to the models and prompting "Please generate a video (or an image in the case of GPT-4o which cannot generate video) similar to the semantics in the input." X-VILA demonstrates significant improvements in visual correspondence over previous methods, thanks to the integration of the Visual Embedding Highway (VEH) into output diffusion models.

**Emergent X-to-X ability.** During our experiments, we observe highly promising emergent abilities displayed by X-VILA following its training on our X-to-X datasets. As depicted in Figure 4, we have identified two key capabilities that have surfaced:
(i) **Long-context cross-modality generation.** X-VILA exhibits an impressive capacity for comprehending and combining diverse concepts from multiple iterations of input. Consequently, it produces natural and coherent output, as suggested by the users.
(ii) **Unseen cross-modality ability.** Remarkably, X-VILA showcases the ability to perform image-to-audio and audio-to-image tasks without any explicit training on similar data. This newfound competence emerges organically through the model's exposure to our comprehensive X-to-X dataset. These remarkable emergent abilities underscore the efficacy of our meticulously curated X-to-X dataset. Not only does it enable the model to excel in the specified data types as suggested in Section 3.2, but it also facilitates generalization across a wide range of multi-modality interactions between users and the model.

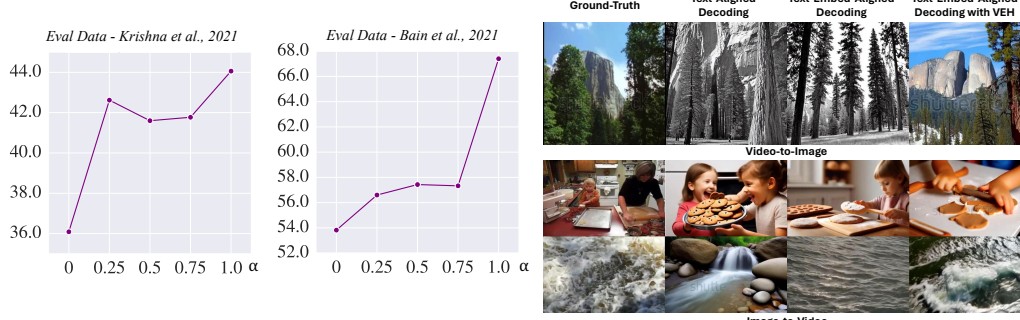

Figure 8: *(left, middle)* X$^2$A scores when using different conditioning rates $\alpha$ in VEH (image) on evaluation data generated from (Krishna et al., 2017) and (Bain et al., 2021). Higher conditioning rates brings generally better X-to-X alignment. *(right)* An in-depth comparison of varying design choices of X-VILA on cross-modality alignment tasks. We observe that both Text-Aligned Decoding and Text-Embed-Aligned Decoding fall short in effectively capturing semantic details from visual inputs. However, with the incorporation of our Visual Embedding Highway (VEH), we witness a substantial improvement in visual consistency.

**More insights on varying design choices on decoder alignment.** We next present our findings when aligning LLM output end to the modality-specific decoders. We study different ways to bridge LLM output and the diffusion models: *(i) "Text-Aligned Decoding"*: LLM generates text description for the expected image/video/audio predictions and then feeds the text description into pre-trained image/video/audio decoders. *(ii) "Text-Embed-Aligned Decoding"*: LLM generates modality-specific generation tokens and then we use the corresponding high-dimensional textual embeddings to control the modality-specific decoders (as described in Section 2.1). *(iii) "Text-Embed-Aligned Decoding with VEH"*: Building upon method (ii), we introduce the Visual Embedding Highway (VEH) to align the visual feature between encoders and decoders. We conduct experiments on video-to-image and image-to-video cross-modality alignment tasks, and show the results on the right side of Figure 8.

The findings suggest that conveying specific details such as visual style, object appearance, and precise human actions from the input to the output is challenging for Text-Aligned Decoding. This difficulty arises due to the low-dimensional nature of pure text descriptions, which limits the amount of information they can contain. On the other hand, Text-Embed-Aligned Decoding offers a significantly greater "bandwidth" in the textual embedding space between the LLM and modality-specific decoders. Consequently, Text-Embed-Aligned Decoding is capable of generating more consistent outcomes. Nevertheless, Text-Embed-Aligned Decoding alone is still not good enough for capturing visual details, as a substantial amount of visual information is lost during the projection from encoders to the LLM. This is where our Visual Embedding Highway demonstrates its performance and aids X-VILA in attaining notably enhanced visual consistency.

**Conversation examples.** To thoroughly investigate the performance of our any-to-any modality LLM, we conducted extensive testing on X-VILA examining many use cases. We present conversation examples of X-VILA across varying tasks in Figure 1 and Figure 9. It can be observed that X-VILA provides users with a comprehensive set of multi-modality responses leveraging the encoders for perception, LLM for understanding and reasoning, and decoders for multi-modality content generation. As shown in Figure 14, X-VILA not only exhibits its understanding of the visual input, including the scene and objects, but also predicts the actions of the person depicted in the image. This capability is a result of training on our extensive X-to-X dataset. Based on the visual input, it generates outputs visually consistent with the input, *e.g.*, the snow mountain and red ski suit are presented in the generation output correctly.

## 4  RELATED WORK

The era of *Large Language Models (LLM)* arguably started with the introduction of transformers (Vaswani et al., 2017) and a series of works that scaled them. Particularly, OpenAI introduced the Generative Pre-trained Transformer (GPT) models (Radford et al., 2019), (Brown et al., 2020),

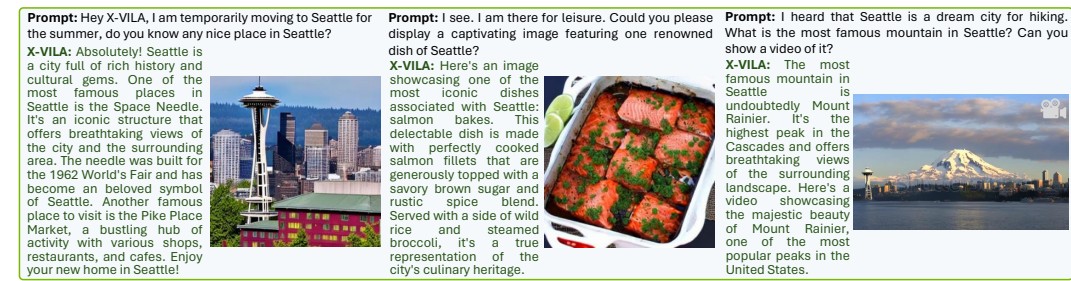

**Prompt:** Hey X-VILA, I am temporarily moving to Seattle for the summer, do you know any nice place in Seattle?

**X-VILA:** Absolutely! Seattle is a city full of rich history and cultural gems. One of the most famous places in Seattle is the Space Needle. It's an iconic structure that offers breathtaking views of the city and the surrounding area. The needle was built for the 1962 World's Fair and has become an beloved symbol of Seattle. Another famous place to visit is the Pike Place Market, a bustling hub of activity with various shops, restaurants, and cafes. Enjoy your new home in Seattle!

**Prompt:** I see. I am there for leisure. Could you please display a captivating image featuring one renowned dish of Seattle?

**X-VILA:** Here's an image showcasing one of the most iconic dishes associated with Seattle: salmon bakes. This delectable dish is made with perfectly cooked salmon fillets that are generously topped with a savory brown sugar and rustic spice blend. Served with a side of wild rice and steamed broccoli, it's a true representation of the city's culinary heritage.

**Prompt:** I heard that Seattle is a dream city for hiking. What is the most famous mountain in Seattle? Can you show a video of it?

**X-VILA:** The most famous mountain in Seattle is undoubtedly Mount Rainier. It's the highest peak in the Cascades and offers breathtaking views of the surrounding landscape. Here's a video showcasing the majestic beauty of Mount Rainier, one of the most popular peaks in the United States.

Figure 9: Examples of X-VILA performing a multi-turn any-to-any modality conversation. Prompts are given left to right in a multi-round manner. Best viewed in color.

from GPT-2 (1.5B parameters) to GPT-4 (OpenAI, 2023a) (1.76T), and showed that parameter scaling, together with more high-quality data, can generate coherent and contextually relevant text across various domains. BERT (Devlin et al., 2019) introduced a paradigm of bidirectional text processing enabling stronger context understanding and boosted question answering. T5 (Raffel et al., 2020) converted language problem into a text-to-text format advancing translation and summarizing. Transformer-XL (Dai et al., 2019) demonstrated the capability of extending the context window allowing for a better understanding of longer text. The application era of LLM was kickstarted by ChatGPT (OpenAI, 2023b) which showcased the unprecedented ability of LLM chatbots.

Current *Vision-Language Models (VLM)* benefited from the development of ViT (Dosovitskiy et al., 2021) that offers a unified way for vision models to communicate with other transformers from different modalities. Rapid progress has been shown in three streams (Awais et al., 2023): **(i)** textually prompted models that accept image and text as input (CLIP (Radford et al., 2021), Frozen (Tsimpoukelli et al., 2021), BLIP (Li et al., 2023a), PaLI (Chen et al., 2023), LLaVa (Liu et al., 2024), VILA (Lin et al., 2024), miniGPT4 (Zhu et al., 2023a)); **(ii)** visually prompted models (CLIPSeg (Lüddecke, 2021), SAM (Kirillov et al., 2023)); and **(iii)** multi-modal input-output models (Painter (Wang et al., 2022), ImageBind (Girdhar et al., 2023), Palm-E (Driess et al., 2023a), Video ChatGPT (Maaz et al., 2023), RegionGPT (Guo et al., 2024), mPLUG-owl (Ye et al., 2023), PandaGPT (Su et al., 2023), CoDi (Tang et al., 2023), NextGPT (Wu et al., 2023), Unified-IO (Lu et al., 2022; 2023)). Among the first, Frozen (Tsimpoukelli et al., 2021) demonstrated that VLM can be constructed by linear projection of ViT features into LLM and only tuning ViT on image-text captioning data. They are the first that discover the few-shot capabilities of VLM without instruction. Flamingo (Alayrac et al., 2022) used cross-attention for vision language binding, and for a first time demonstrated surpassing state-of-the-art finetuned models for multiple tasks. PALI (Chen et al., 2023) created a universal model that can do vision and language tasks separately, they scaled ViT to 4B and demonstrated the importance of adding language-only data to the pretraining stage. Overall, VLM follows the pipeline of taking a pretrained LLM; adding a pretrained vision encoder; learning feature alignment at scale via a projector or cross-attention; followed by instruct-tuning (InstructBLIP (Dai et al., 2023), FLAN (Wei et al., 2021)). In close relation to our research, Next-GPT introduces an LLM that possesses the capability to comprehend multi-modality inputs and generate corresponding multi-modality outputs through textual alignment, yet it cannot effectively handle visual details present in the input.

## 5 CONCLUSION

This paper presents X-VILA, an any-to-any modality LLM that is able to understand, infer, and generate multi-modality contents. This ability is achieved through any-to-any modality alignment, for which we curate a dataset for any-to-any modality instruction tuning. We further identify a significant drawback in the previous textual alignment method that leads to the loss of crucial visual details. Accordingly, we propose an innovative visual alignment mechanism that incorporates a visual feature highway module. This solution helps preserve essential visual details from the input. The experimental results, both quantitative and qualitative, indicate the effectiveness of our data and methodology. X-VILA's performance can be further enhanced across various VLM benchmarks.

## ETHICS STATEMENT

Our proposed method does not involve the creation or introduction of any new image/video/audio content other than open sourced datasets used by prior academic work. All data used in this project is intended exclusively for academic research purposes and will not be used for any commercial applications.

## REPRODUCIBILITY STATEMENT

The project will be open-source to help the research community to reproduce. We elaborate on our model design in Section 2.1. Additionally, we outline the training and implementation details, including the training hyperparameters in Section A and C in the appendix.

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

# A  X-VILA TRAINING

The training process of X-VILA is divided into three phases, namely (i) encoder-LLM-Decoder alignment training, (ii) interleaved data pre-training, and (iii) X-to-X cross-modality instruction fine-tuning.

## A.1  ENCODER-LLM-DECODER ALIGNMENT TRAINING PHASE.

As the first step, we align the output of modality-specific encoders and the input of modality-specific decoders to the textual embedding space of LLM, as detailed in Wu et al. (2023). To achieve this goal, we only train the input projection layers, output projection layers, and the vocabulary embedding layer of LLM, while keeping all other parameters frozen. We use corpus with "X"-text pairs to train the model, where "X" is one of the video, image, or audio modalities.

For this stage, we design two primary tasks to train the projection layers: X-to-text generation and text-to-X generation.

**(a)** X-to-text generation includes video, image, and audio captioning tasks. The model is supervised to generate text based on the multi-modality inputs. During this process, the input projection layers are trained to align the output embedding of modality-specific encoders and the textual embedding space of pre-trained LLM.

**(b)** Text-to-X generation aims at aligning the output textual embedding space of LLM and the input end of modality-specific decoders. We use video, image, and audio generation tasks to train the model, where only the output projection layers are optimized. As previously mentioned, the training objective here is pure textual alignment: minimizing the feature distance between the textual controller embedding $\mathbf{E}_m^{\text{text}}$ generated by the output projection layers and the embedding generated by the original pre-trained text encoder of diffusion model. This training strategy ensures that $\mathbf{E}_m^{\text{text}}$ shares a distribution similar to that of the pre-trained text encoder in the diffusion model. After training, $\mathbf{E}_m^{\text{text}}$ replaces the diffusion text encoder feature to control the U-Nets of the modality-specific decoders via cross-attention.

## A.2  INTERLEAVED DATA PRE-TRAINING PHASE.

Interleaved data training has been proven to be an effective strategy for vision-language models in alleviating the catastrophic forgetting issue after training on only visual-text pairs, and obtaining long-context understanding ability Lin et al. (2024); Awadalla et al. (2023). Therefore, we introduce a dedicated phase for pre-training X-VILA using a multi-modality interleaved corpus.

In addition to interleaved image-text pairs as in MMC4 Zhu et al. (2023b), we further construct a new dataset from ActivityNet Captions Krishna et al. (2017). The main idea is to exploit the nature of video that contains sequential flow of text (*e.g.*, captions), audio, short video, and image. This enables us to put the images/videos and texts in an interleaved manner, and use the corpus to pre-train X-VILA.

Specifically, we construct interleaved multi-modality data sequences from each target video clip as:

$$\underbrace{\{\texttt{, <aud. 1>, <vid. 1>, <txt 1>}\}}_{\text{sampled from video chunk 1}}, ..., \underbrace{\{\texttt{, <aud. n>, <vid. n>, <txt n>}\}}_{\text{sampled from video chunk } n},$$

where the video chunks are sampled from an entire video clip that offers natural sources of interleaved cross-modality data structure. Once constructed, the modalities are sampled during training to align varying targets for gradient computation and network projector alignment. In this work, we observe the even sampling method and $n = 3$ are sufficient for the task, namely constructing cross-modality tasks for the beginning, middle stage, and ending of video clips. During this stage, we jointly train the input and output projection layers, and use LoRA Hu et al. (2021) on LLM for fine-tuning.

### A.3 X-TO-X CROSS-MODALITY INSTRUCTION TUNING PHASE.

After the previous two phases, we have textually aligned different components of X-VILA in a unified framework. However, the model is still not ready for understanding and generating multi-modality content in a proper manner. To achieve this goal, we curate a comprehensive "X-to-X dataset" for cross-modality generation instruction tuning. Our X-to-X dataset features six different types of cross-modality generative conversations, namely **video-to-image, video-to-video, image-to-video, video-to-audio, audio-to-video, and image+audio-to-video**. We show examples of different types of conversations in Figure 10. Each conversation contains one or more rounds of cross-modality conversation. More details about the X-to-X dataset are described in the experiment section.

We further divide the X-to-X cross-modality instruction tuning phase into two distinct steps, each based on different alignment methods: textual alignment and visual alignment.

**(a)** To achieve textual alignment, we first project the multi-modality inputs into the textual embedding space of LLM. Then, LLM generates textual embeddings that are subsequently converted into the corresponding modality's content. We follow a process similar to phases (i) and (ii). Firstly, for image, video, or audio outputs, we generate embeddings using the text encoders of corresponding diffusion models. We then optimize the distance between these embeddings and the $\mathbf{E}_m^{\text{text}}$ generated by our model. During this step, we keep all the decoder weights frozen and train the input projection layers, output projection layers, and vocabulary embedding layer as well as LoRA parameters of LLM. For training data, we blend our X-to-X dataset with common SFT datasets used by other VLM models Liu et al. (2024); Wu et al. (2023) (more details in the experiment section).

**(b)** As mentioned earlier, relying solely on textual alignment is inherently insufficient to retain the visual details of the input when generating visual outputs. To address such an issue, we design a novel visual alignment method. We propose a visual embedding highway (VEH) module as introduced in Section 2.1, which is utilized for the image and video decoders when there is a visual modality in the input. During training, we update the parameters of the visual decoders and the visual controller module. Meanwhile, we keep all other network parameters fixed, including the input and output projection layers and LLM. In this way, the model's ability to conduct tasks in other modalities is not influenced by the visual alignment process.

## B MORE QUALITATIVE RESULTS

### B.1 EXAMPLES OF OUR X-TO-X DATASET.

To provide an intuitive understanding of the six types of conversations in our curated X-to-X dataset, we visualize the conversation samples of the dataset in Figure 10. The design of the dataset focuses on building any-to-any modality connection through various conversation templates.

### B.2 HUMAN-MODEL INTERACTION DEMONSTRATION.

To conduct a comprehensive assessment of our any-to-any modality LLM's performance, we undertake more testing on X-VILA, meticulously examining different use cases. We present a collection of human-model conversation examples in Figure 11, 12, 13 and 14, showcasing the versatility of X-VILA across diverse tasks. These results demonstrate the effectiveness of X-VILA in addressing the needs of users by offering comprehensive and generative multi-modality capabilities.

## C MORE IMPLEMENTATION DETAILS

As introduced in Section A, X-VILA training is separated into three phases. (i) In the initial phase, referred to as encoder-LLM-decoder alignment training, the model undergoes 20,000 iterations using an Adam optimizer. The base learning rate is set to $4 \times 10^{-4}$, and a learning rate warm-up strategy is employed. The batch size for this phase is set to 200. (ii) During the second phase, known as interleaved data pre-training, a batch size of 192 is utilized. The base learning rate is set to $1 \times 10^{-4}$, and the training is conducted for 10,000 iterations. (iii) The final phase, called cross-modality instruction tuning, involves separate training for textual and visual alignment. For textual

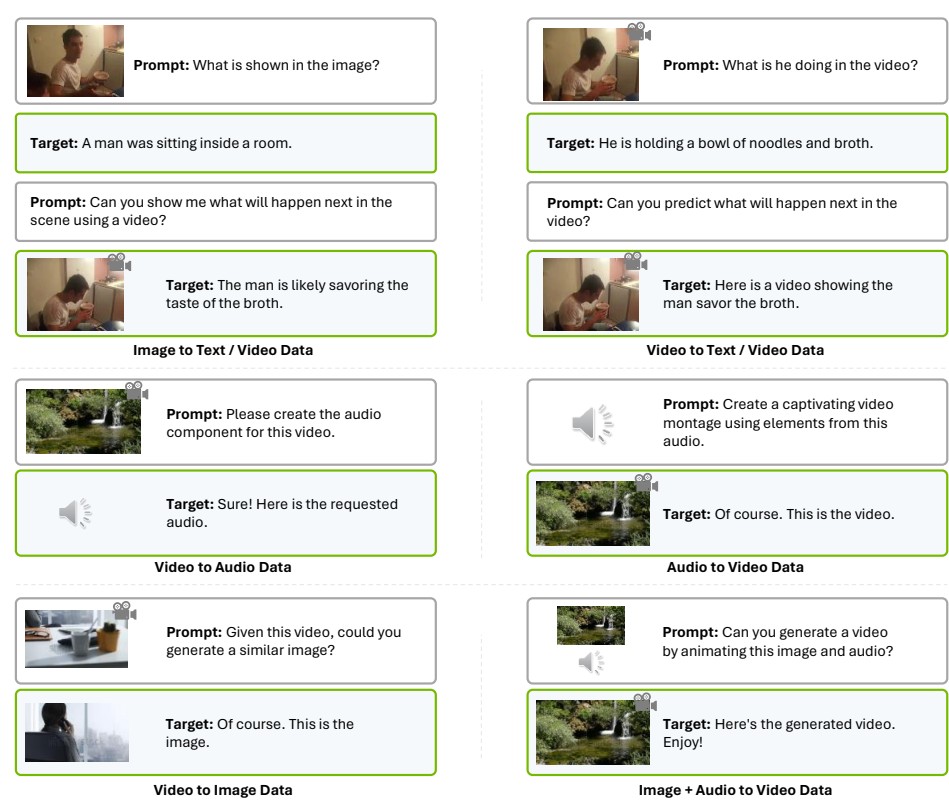

Figure 10: Examples of different types of conversations in our X-to-X dataset. They cover six types of cross-modality understanding and generation tasks.

alignment, a batch size of 192 is maintained, and the model is trained for 30,000 iterations using a base learning rate of $1 \times 10^{-4}$. Regarding visual alignment, both the Visual Embedding Highway (VEH) and modality-specific decoders are trained for 20,000 iterations. The batch size for this phase is set to 64, and the learning rate is adjusted to $1 \times 10^{-6}$. For data amount, Our training pipeline is highly efficient compared to many previous vision-language models Alayrac et al. (2022); Bai et al. (2023b); Chen et al. (2023). We utilize a total of 4 NVIDIA A100 80GB server nodes in the training process.

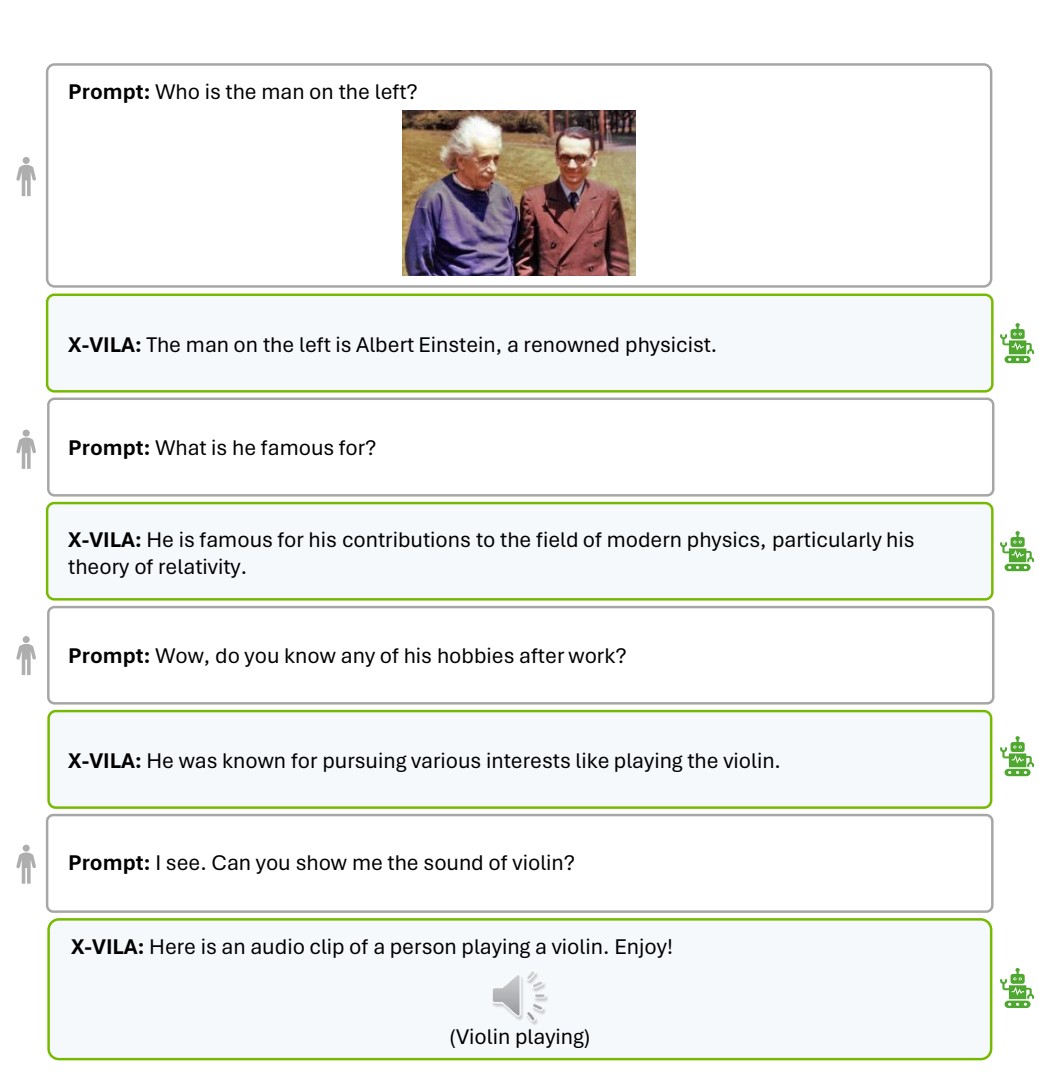

Figure 11: Example of X-VILA performing cross-modality chat. X-VILA shows strong multi-understanding and generation ability.

**Prompt:** I am going to travel in Paris, how does it look like?

**X-VILA:** Here's a video showcasing the beauty of Paris, the city of romance. You'll get to see its iconic landmarks, charming cafes, and the vibrant atmosphere that makes it so special. Enjoy the virtual tour!

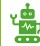

**Prompt:** Does Paris have any special music genre?

**X-VILA:** Absolutely! Paris is known for its diverse and vibrant music culture. Some of the most popular styles include French pop, jazz, and classical music. You can also find a wide range of modern and experimental musicians in the city's bustling music scenes.

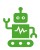

**Prompt:** Wow, can you show me an audio of French Jazz?

**X-VILA:** Sure! Here's an audio clip that captures the lively and vibrant sounds of French Jazz. Enjoy the rhythmic beats and harmonious melodies that make this genre so enjoyable.

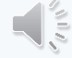

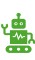

Figure 12: Example of X-VILA performing cross-modality chat. X-VILA shows strong multi-understanding and generation ability.

**Prompt:** The summer is coming, where do you think is a good place for summer vacation?

**X-VILA:** There are many great places to visit for a summer vacation. Some popular options include beaches, national parks, and mountain resorts. Beaches offer a chance to relax and soak up the sun, while national parks provide opportunities for hiking, camping, and exploring nature. Mountain resorts offer a range of outdoor activities such as golfing, hiking, and climbing, as well as a variety of cultural events and attractions. Ultimately, the best destination for your summer vacation will depend on your personal interests and travel preferences.

**Prompt:** Thanks. I would like to go to the beach, do you have any recommendations? Please show me a video.

**X-VILA:** Certainly! There are many beautiful beaches around the world that would make for a great summer vacation. Here is a video showcasing some of them. Enjoy!

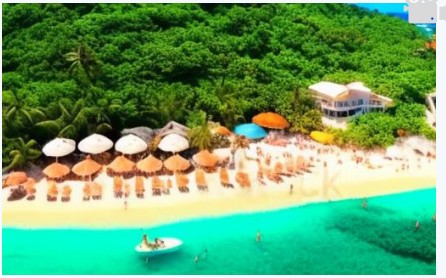

**Prompt:** Cool. I am actually a businessman. What do you think sell well on the beach? Can you show an image example?

**X-VILA:** As a businessman, you may be interested in selling products or services that cater to the needs and desires of beachgoers during their summer vacation. Here is an image of a delicious summer picnic on the beach, complete with a variety of fresh and delicious food items like fruits, vegetables, sandwiches, and more.

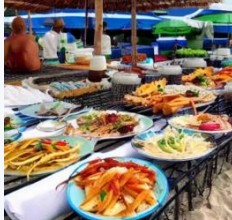

Figure 13: More examples of X-VILA performing cross-modality chatting.

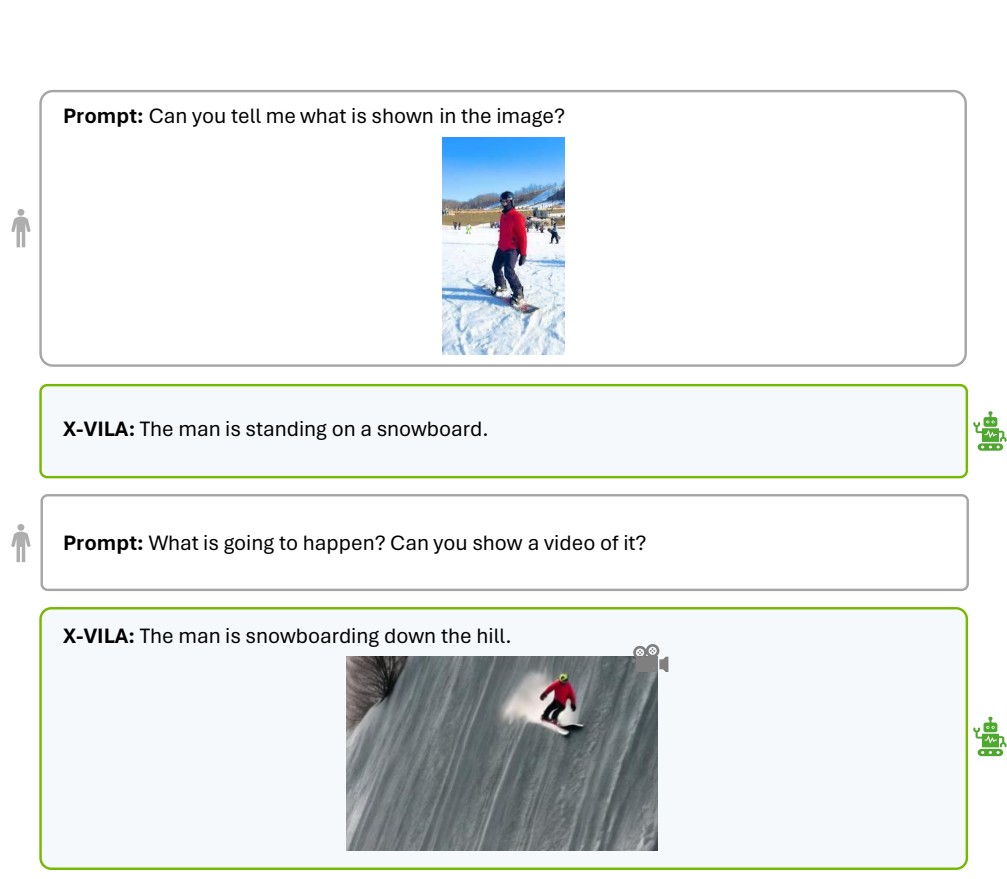

Figure 14: Conversation examples of X-VILA performing cross-modality conversation. X-VILA demonstrates a remarkable capability to comprehend the visual input and perform reasoning based on it. Our proposed visual alignment technique plays a crucial role in establishing visual consistency between the input and output.

