# OpenReview forum: "X-VILA: Cross-Modality Alignment for Large Language Models"
_ICLR.cc/2025/Conference — ICLR 2025 Conference Withdrawn Submission_

### Official Review · Reviewer_giZL · 2024-11-01

**Soundness:** 2
**Presentation:** 3
**Contribution:** 2
**Rating:** 5
**Confidence:** 4

**Summary:**

The paper introduces X-VILA, an omni-modality model designed to extend the capabilities of large language models (LLMs) by incorporating image, video, and audio modalities. X-VILA achieves cross-modality understanding, reasoning, and generation by aligning modality-specific encoders with LLM inputs and diffusion decoders with LLM outputs. The key contributions of the paper are:

- It presents a new family of any-to-any modality chat LLMs capable of conducting multi-modality conversations, understanding signals from different modalities, and generating content in various formats, including video, audio, image, and text.
- The paper introduces a new X-to-X multi-modality instruction tuning dataset, which has proven effective for cross-modality alignment.
- To address the issue of visual information loss, a visual alignment mechanism with a visual embedding highway module is introduced, which allows visual features to bypass the LLM, enhancing the correspondence of visual content between input and output stages.
- X-VILA demonstrates emergent properties across modalities even in the absence of similar training data, showcasing abilities like long-context cross-modality generation and new types of cross-modality ability.

**Strengths:**

The introduction of the Visual Embedding Highway (VEH) module to preserve visual details is an original contribution that enhances the model's performance in visual tasks.

The paper is of high quality, as evidenced by its thorough methodology, comprehensive experiments, and rigorous evaluation.

The paper is well-structured and clearly written.

**Weaknesses:**

- How do the authors ensure the data quality of the proposed X-to-X dataset, and what is the specific process for its creation?

- Why is the additional module VEH implemented solely in the visual component?

- Compared to Next-GPT, the method in this paper utilizes a larger instruction tuning dataset. Could the authors provide experimental results demonstrating the impact of scaling up the instruction data?

**Questions:**

Please refer to the weaknesses section.

---

### Official Review · Reviewer_vPJC · 2024-11-03

**Soundness:** 3
**Presentation:** 3
**Contribution:** 3
**Rating:** 8
**Confidence:** 3

**Summary:**

The paper introduces X-VILA, a model capable of handling various input and output modalities, including video, image, language, and audio. X-VILA employs a two-step alignment mechanism involving textual alignment and a novel visual alignment component called the Visual Embedding Highway (VEH). This mechanism addresses common challenges in multi-modality alignment, such as loss of visual information. The authors contribute a large-scale X-to-X multi-modal instruction dataset to support further research in multi-modal models.

**Strengths:**

1. The introduction of the Visual Embedding Highway (VEH) to preserve and enhance visual features represents a significant advancement over existing methods in visual alignment.
2. The paper provides a large X-to-X dataset for multi-modal instruction, which is a valuable resource for advancing research in multi-modal modeling.
3. The authors plan to open-source X-VILA and the dataset, promoting academic and engineering research in multi-modal foundation models.

**Weaknesses:**

1. Missing Example Component: The example in Figure 1 lacks the "video" input component, which disrupts the clarity and completeness of the illustration.
2. High Computational Requirements: The model’s training process is resource-intensive, potentially limiting accessibility and reproducibility, particularly for smaller research teams.

**Questions:**

Is there a planned timeline for releasing the training code and datasets for X-VILA?

---

### Official Review · Reviewer_vhae · 2024-11-03

**Soundness:** 2
**Presentation:** 2
**Contribution:** 2
**Rating:** 3
**Confidence:** 5

**Summary:**

This paper propose a multi-modality large language model which can model the understanding and generation between audio, image, video and text. It also propose a X-to-X generation instructional tuning dataset. The method is evaluated on multimodal generation benchmark and some general visual/audio QA/caption benchmarks.

**Strengths:**

1.The instructional tuning dataset may be helpful for multi-modality generation research.
2. Proposed method surpasses Next-GPT on compared benchmarks
3. Proposed Visual Embedding Highway sounds make sense.

**Weaknesses:**

1. Novelty is limited. Unifing text and visual generation have been studied for a lot, such as Cogview or Emu series.  Adding audio modality also have been researched by AnyGPT or NextGPT. So this paper seems more like some performance incresements istead of some new insights, which is not excited enough to me.
2. Architecture design does not have new insights. Multimodal encoders + projectors + LLM + diffusion decoders are a  common way for unifying multimodal understanding and generation. Even though authors  emphasize that diffusion models are tuned together which make difference from earlier works, but the architecture still seem not novel and elegant enough. In addition, the visual high way design seems serving for edit-like tasks targetedly instead of a general design.
3. Performances on visual QA or audio caption benchmarks are too weak.

**Questions:**

1. Why use Imagebind as all modality encoders? It seems all tasks performs bad, If we change "unbind encoders", such as clip for vision, whisper/beats for audio, the performance could definitely be better. Imagebind only sounds better instead of performance better.
2. Number of compared methods are too small. Only NextGPT?
3. " As depicted in Figure 4, we have identified two key capabilities that have surfaced Long-context cross-modality generation. X-VILA exhibits an impressive capacity for comprehending and combining diverse concepts from multiple iterations of input. Consequently, it produces natural and coherent output, as suggested by the users."  I don't think the context is long enough to be called "long-context" in the example of Figure 4.
4. The first stage training, so-called "encoder-LLM-Decoder alignment training" utilizes the "X-text pairs" from academic datasets as in prior work, so what does the decoder train? It is so confused.

---

### Official Review · Reviewer_o9od · 2024-11-04

**Soundness:** 2
**Presentation:** 2
**Contribution:** 2
**Rating:** 3
**Confidence:** 4

**Summary:**

This paper introduc X-VILA, an omni-modality model designed to extend the capabilities of large language models by incorporating image, video, and audio modalities.  X-VILA achieves cross-modality understanding, reasoning, and generation within one model. And related capabilities are evaluated on extensive benchmarks.

**Strengths:**

This paper represents a commendable attempt in the field of multimodal integration. The proposed Vehicle for Enhancing Harmony (VEH) offers valuable insights into facilitating the interaction between perception modality encoding and decoding. Overall, the manuscript is well-written and provides a thorough performance evaluation.

**Weaknesses:**

* **Confusion in Structural Design**: According to the authors, the Embedding Highway design is intended to be applicable across various modalities. However, it is only applied to the visual modality, which appears to contradict the authors' objective of constructing a truly Any-to-Any multimodal model.

* **Novelty of the Model**: To my knowledge, the architecture that employs modality-independent encoding and decoding, integrated through a LLM, has been extensively explored in several prior works (e.g., Next-GPT [1]). This architecture does not seem to demonstrate clear novelty. It would be beneficial for the authors to systematically compare their work with these existing studies to highlight its innovative aspects.

[1] Wu S, Fei H, Qu L, et al. NExT-GPT: Any-to-Any Multimodal LLM[C]//Forty-first International Conference on Machine Learning.

**Questions:**

* I personally appreciate the architecture that supports multimodal perception and generation. However, it is true that vision and audio alone do not fully encompass the concept of Any-to-Any; other modalities are also prevalent. It appears that the current architecture lacks sufficient scalability for additional modalities. Introducing new modalities often requires re-training the LLM, which poses a limitation that may not be conducive to building a truly Any-to-Any architecture.

* I would like to understand the nature of the interactions between modalities. For instance, how does modeling and learning within the audio modality affect that of the image modality?

---

### Note · Authors · 2024-11-13

**Comment:**

We would like to sincerely thank all reviewers for their valuable feedback, and we have decided to withdraw the submission for further improvement.

**Withdrawal Confirmation:**

I have read and agree with the venue's withdrawal policy on behalf of myself and my co-authors.